# Cysteine-Mediated Extracellular Electron Transfer of *Lysinibacillus varians* GY32

Guannan Kong,[a] Yonggang Yang,[a] Yeshen Luo,[a] Fei Liu,[a] Da Song,[a] Guoping Sun,[a] Daobo Li,[a] Jun Guo,[a] Meijun Dong,[a] Meiying Xu[a]

[a]Guangdong Provincial Key Laboratory of Microbial Culture Collection and Application, State Key Laboratory of Applied Microbiology Southern China, Institute of Microbiology, Guangdong Academy of Sciences, Guangzhou, China

**ABSTRACT** Microbial extracellular electron transfer (EET) is essential in many natural and engineering processes. Compared with the versatile EET pathways of Gram-negative bacteria, the EET of Gram-positive bacteria has been studied much less and is mainly limited to the flavin-mediated pathway. Here, we investigate the EET pathway of a Gram-positive filamentous bacterium *Lysinibacillus varians* GY32. Strain GY32 has a wide electron donor spectrum (including lactate, acetate, formate, and some amino acids) in electrode respiration. Transcriptomic, proteomic, and electrochemical analyses show that the electrode respiration of GY32 mainly depends on electron mediators, and *c*-type cytochromes may be involved in its respiration. Fluorescent sensor and electrochemical analyses demonstrate that strain GY32 can secrete cysteine and flavins. Cysteine added shortly after inoculation into microbial fuel cells accelerated EET, showing cysteine is a new endogenous electron mediator of Gram-positive bacteria, which provides novel information to understand the EET networks in natural environments.

**IMPORTANCE** Extracellular electron transport (EET) is a key driving force in biogeochemical element cycles and microbial chemical-electrical-optical energy conversion on the Earth. Gram-positive bacteria are ubiquitous and even dominant in EET-enriched environments. However, attention and knowledge of their EET pathways are largely lacking. Gram-positive bacterium *Lysinibacillus varians* GY32 has extremely long cells (>1 mm) and conductive nanowires, promising a unique and enormous role in the microenvironments where it lives. Its capability to secrete cysteine renders it not only an EET pathway to respire and survive, but also an electrochemical strategy to connect and shape the ambient microbial community at a millimeter scale. Moreover, its incapability of using flavins as an electron mediator suggests that the common electron mediator is species-dependent. Therefore, our results are important to understanding the EET networks in natural and engineering processes.

**KEYWORDS** cysteine, Gram-positive bacteria, extracellular electron transfer, transcriptome, proteome

xtracellular electron transfer (EET) is a process by which microbes transfer electrons through the cell envelope to electron acceptors outside the cell (1, 2). It plays a key role in various natural environments and bioelectrochemical systems for bioenergy, resource recovery, and wastewater treatment (3–5). In recent years, EET mechanisms have been extensively studied, mainly with Gram-negative bacteria such as *Geobacter* and *Shewanella* (1, 6, 7). Two mechanisms of EET have been proposed: (i) direct contact between bacterial cells and extracellular electron acceptors via *c*-type cytochromes or nanowires (6–8), and (ii) indirect EET via dissolved electron mediators (e.g., flavins or phenazines), or metabolites (e.g., $H_2$ or formate) in interspecies electron transfer (9–11). The EET of Gram-positive bacteria has caught little attention, despite the fact that Gram-positive bacteria are also widespread or even dominant in various EET-driven processes (12). Gram-positive bacteria have a thick layer of peptidoglycan (typically 30 to 100 nm) in the cell wall, and their cell wall is significantly thicker than

Address correspondence to Yonggang Yang, yyg117@163.com, or Meiying Xu, xumy@gdim.cn.

The authors declare no conflict of interest.

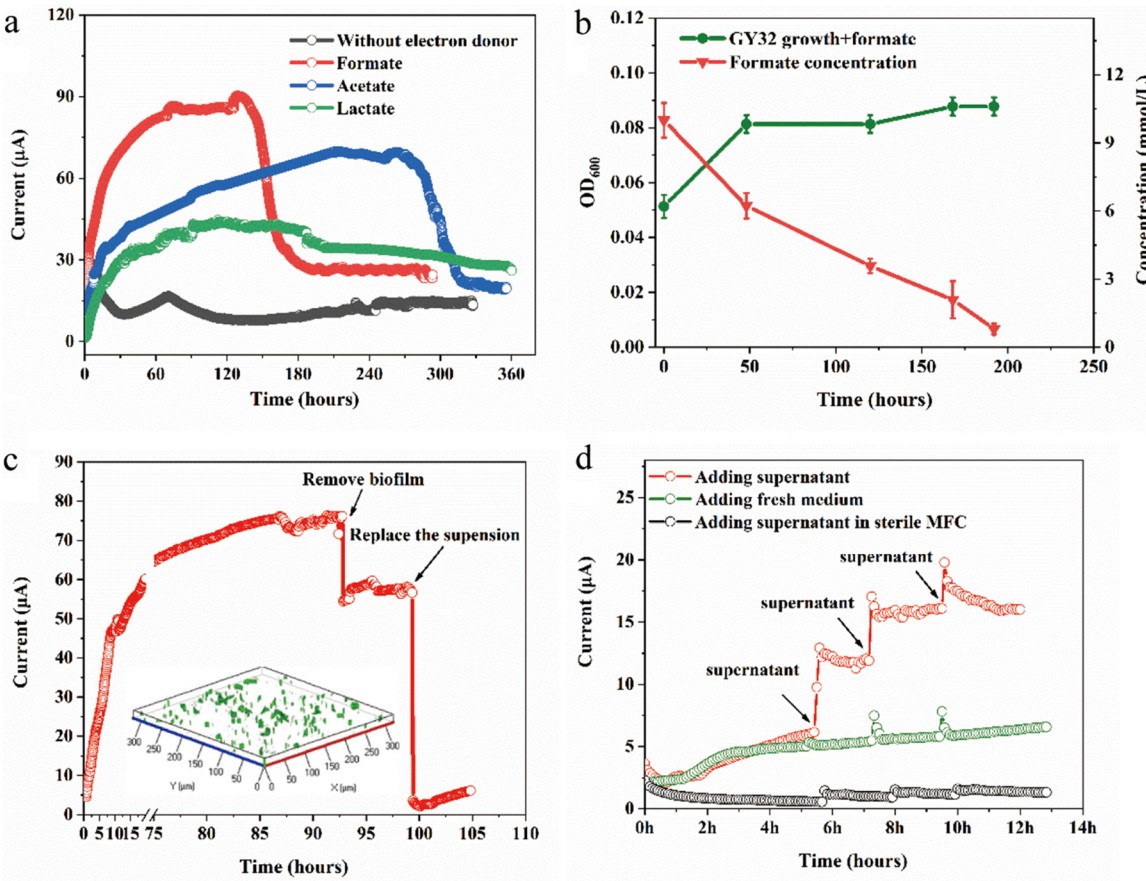

**FIG 1** Current generation and cell growth of strain GY32 in MFC. (a) The current output of GY32 using different electron donors. (b) The growth of strain GY32 and depletion of formate in MFC. (c) Contributions of different components in anodic chamber to MFC current, insert shows the anode biofilm (300 × 300 $\mu$m). (d) Effects of adding culture supernate on current generation.

Gram-negative bacteria (typically 2 to 7 nm). The thicker cell wall of Gram-positive bacteria had been considered an obstacle for EET (13, 14). However, the cross-linked structure of peptidoglycan of Gram-positive bacteria contains numerous pores with a diameter of 50 to 500 Å, allowing many *c*-type cytochromes and electron mediators to pass through (15). Consistently, transmembrane *c*-type cytochrome chain (16), nanowires (17), and electron mediators (18–20) have been increasingly reported in Gram-positive bacteria. Among them, mediators were considered the most common EET pathway. To date, EET mediators generated by Gram-positive bacteria are still limited to flavins, and other kinds of mediators for EET have not been reported yet (19, 20).

*Lysinibacillus varians* GY32 is a filamentous Gram-positive bacterium with an extremely long unicellular cell shape (over 1 mm). It can perform bidirectional EET and form centimeter-scale conductive cellular networks when respiring with graphite electrodes (17). The unique structural and electrochemical properties indicate that *L. varians* GY32 have special performances in the interactions with electrodes or neighbor microorganisms (17). However, the available substrates and EET pathways remain unclear. In this study, the EET of *L. varians* GY32 with electrodes as the electron acceptor was comprehensively characterized, including the available electron donors, EET pathways, transcriptome, and proteome characteristics. The results reveal that endogenous electron mediators, typically cysteine, play a key role in the EET of *L. varians* GY32.

## RESULTS AND DISCUSSION

**Available electron donors and current generation characteristics of GY32.** *L. varians* GY32 is an electroactive microorganism isolated from polluted river sediment (17). Fig. 1a shows the current generation by strain GY32 when three naturally ubiquitous electron donors,

formate, acetate, and lactate, were added to the microbial fuel cells (MFC) separately (21). Compared with the background current and acetate-/lactate-fueled current, the formate-fueled MFC showed the most rapid and highest current generation (90.3 $\mu$A), demonstrating that GY32 could perform more rapid EET to electrode when it used formate as an electron donor. In addition, amino acids are also ubiquitous possible electron donors for microbial electrode respiration in environments (22, 23). We tested 20 types of amino acid as electron donors, and the result demonstrated that strain GY32 could use five (L-glutamate, L-asparagine, L-alanine, L-histidine, L-serine) of them to generate significantly higher current (over 40 $\mu$A) than the control MFC without electron donor (maximum background current of 26 $\mu$A) (Fig. S1). These results indicate that the Gram-positive bacterium *L. varians* GY32 may have a wide electron donor spectrum for EET in natural sediment environments.

The EET in formate-fueled MFC was further studied because: (i) strain GY32 performed relatively high EET rates using formate than the other electron donors; (ii) formate is a prominent microbial fermentation product and is ubiquitous in natural environments; (iii) the cellular proton motive force generated by formate oxidization can stimulate the uptake of other potential carbon sources and microbial growth in anaerobic environments (24). In formate-fueled MFC, the current generation was accompanied by the depletion of formate (Fig. 1b). Although the biomass of GY32 increased slightly during the electrode respiration, no biomass increase was observed in MFCs without yeast extract, indicating that formate served only as an electron donor while the trace yeast extract (0.05%) served as a carbon source for GY32 (Fig. S1d). By comparing the current generation of MFC with or without formate (Fig. S1c), the Coulombic efficiency of the formate-fueled MFC (containing trace minerals but not yeast extract) was calculated to be 18%. The lower Coulombic efficiency were not surprising, as detectable oxygen (up to 0.1 $\mu$M) diffusion into the anode chamber was similar to other experiments with facultative anaerobes (25). Moreover, it has been reported the hydrogen generation caused by hydrogenases of *S. oneidensis* MR-1 could account for 38% of electron loss in the current generation (25). The Ni-Fe hydrogenases of GY32 (WP_025219112, WP_025219113) may also divert electron transfer from current generation to hydrogen production. Although the Coulombic efficiency is lower than that of the bioelectrosystems catalyzed by *Geobacter* species (varied between 30% and 93%) (26), it is comparable with that of the MFCs with *S. oneidensis* MR-1 using formate as the electron donor (19.3%) and higher than some reported Gram-positive bacteria (e.g., 0.96% for *Clostridium cochlearium* and 2.3% for *Bacillus subtilis* RH33) (27–29). Further researches are need to determine the reasons causing the electron loss. Sparse biofilms were observed on the anode surfaces (Fig. 1c), indicating that the EET strategy of GY32 may be indirect rather than contact-dependent, which is more like *Shewanella* species but not *Geobacter* species (30, 31).

We further evaluated the contribution of potential electron mediators to the current generation (10). Fig. 1c shows that the MFC current decreased by 18% when the biofilm-grown anode was replaced with a sterilized fresh anode, and the MFC current decreased by 90% when the culture suspension was replaced with a fresh medium during the plateau current generation state. These results indicate that culture suspension played the main role in MFC current generation. Furthermore, the effect of culture supernatant on MFC current generation is shown in Fig. 1d. The addition of culture supernatant could increase the current generation from 6.4 $\mu$A to 16.2 $\mu$A of biofilm-MFC containing biofilm and fresh medium, while adding fresh medium to the MFC or adding supernatant to a sterile MFC had little effect (<1 $\mu$A) on the current generation (Fig. 1d). These results imply that there are redox chemicals secreted by strain GY32 in the anodic culture, which may play an important role in the EET process from GY32 to the electrode.

**Transcriptomic and proteomic analyses.** To further study the EET mechanism of GY32, we compared the transcriptomes and proteomes of GY32 cells respiring with oxygen or electrode, respectively. In transcriptome analysis, a total of 4,376 genes were detected, among which 1,761 genes were significantly upregulated and 485 genes were significantly downregulated in electrode-respiration compared to oxygen respiration. In proteome analysis, 2,877 proteins were identified, and 146 proteins were significantly upregulated while 121 proteins were significantly downregulated in electrode-respiration compared with oxygen-respiration.

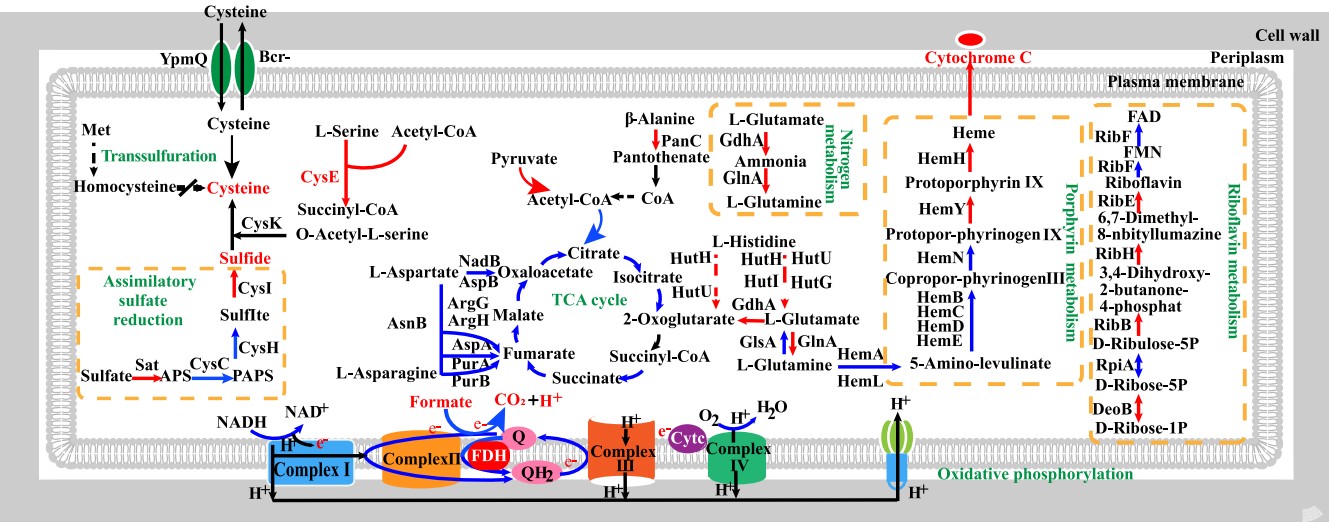

**FIG 2** The enriched KEGG pathways in transcriptome and proteome. The solid blue line represents significant upregulation of gene transcription, the solid red line represents significant upregulation of gene expression, the red dotted line represents significant downregulation of gene expression, the black dotted line indicates that no protein was detected, the black solid line indicates that there is no significant difference in gene transcription and expression.

Combining both the transcriptome and proteome data, 45 metabolic pathways belonging to 12 categories (including amino acid metabolism, carbohydrate metabolism, membrane transport, signal transduction, metabolism of other amino acids, metabolism of cofactors and vitamins, lipid metabolism, glycan biosynthesis and metabolism, folding, sorting and degradation, energy metabolism, cell motility, nucleotide metabolism) showed significant differences (Fig. S2; Table S1).

**General energy and carbon metabolisms.** The transcription and expression of genes that showed significant differences are shown in Fig. 2 and Table S1. For oxidative phosphorylation, when grown in MFCs, the transcriptions of 24 genes were significantly enhanced compared with aerobic growth, while most of them showed no significant difference in the proteome, except for the significantly higher expression of cytochrome $B_5$ (CbaB) in MFC. For the tricarboxylic acid (TCA) cycle, all genes involved in the TCA cycle were expressed under both electrode-respiration and aerobic conditions. Ten of them showed significantly higher transcription level during electrode-respiration but no significant difference was detected in the proteome.

It is worth noting that several amino acids metabolisms showed significant differences. The proteins involved in glutamate degradation (glutamine synthetase GlnA, glutamate dehydrogenase Gdh), glycine degradation (glycine cleavage system protein GcvT), alanine degradation (pantoate–beta-alanine ligase PanC), and arginine degradation (arginase RocF) had higher abundance during the respiration with electrode than oxygen (Table S1). These results indicate that GY32 metabolizes these amino acids at higher rates under electrode respiration, while the cysteine generation increased during current generation (Fig. 2; Fig. 3). The proteins involved in degradation metabolisms of branched-chain amino acids (valine, leucine, and isoleucine) and proline had no significant difference in abundance during the respiration with electrode and oxygen (Table S1). These results are consistent with the availability of amino acids as electron donors by strain GY32 during electrode respiration, i.e., glutamate, glycine, alanine, and arginine could increase the maximum current by at least 10 $\mu$A, while the branched-chain amino acids showed comparable maximum current to the control (Fig. S1). Although strain GY32 could use histidine as the electron donor in electrode respiration, the proteins involved in histidine degradation (histidine ammonialyase HutH, imidazolonepropionase HutI, and formimidoylglutamase HutG) had lower abundance during the respiration with electrode than oxygen (Table S1), probably because histidine was not used for the current generation when there were other electron donors (e.g., formate, glutamate, and asparagine) performing more rapid current generation in MFC (Fig. 1a; Fig. S1a).

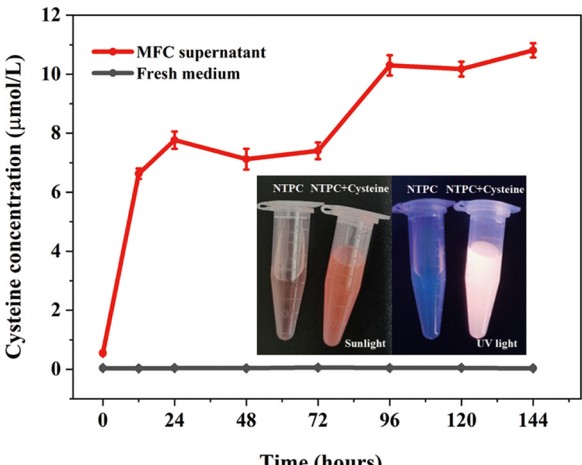

**FIG 3** Cysteine concentration during the electrode respiration of GY32. The inset shows the color and fluorescence changes under sunlight and UV-light before and after cysteine (10 $\mu$M) is added to NTPC, respectively.

**Flavins.** Flavins, mainly flavin mononucleotide (FMN) and riboflavin (RF), are the most common EET mediators in both Gram-negative (e.g., *Shewanella*) and Gram-positive electroactive bacteria (e.g., *Bacillus* and *Listeria*) (13, 19, 20, 32). FMN was detected (up to 1.1 $\mu$M) in the culture liquid of strain GY32 (Fig. 4c), and the concentration is comparable with that of some other reported electroactive bacteria (33). However, the proteins involved in flavin synthesis (RibB, RibH and RibE) had a significantly lower abundance during the respiration with electrode than oxygen (Fig. 2; Table S1). To test the possible role of flavins in the EET of strain GY32, we added flavins (2.0 $\mu$M FMN and RF) to the MFC anodic chamber of GY32 after 40 h of growth and found that they could not increase the current generation (Fig. 4d). These results showed that GY32 synthesized flavins but did not benefit from higher levels after growth for 40 h, probably because the GY32 genome lacks the genes encoding proteins that can reduce and transport flavins, such as the demethylmenaquinone synthase DmkA, DmkB, and lipoprotein PplA in *Listeria monocytogenes* and many other Gram-positive bacteria (19).

**C-type cytochromes.** GY32 contains six putative *c*-type cytochrome genes (17), which contain one or two hemes (Table S2). This is in contrast to the multiheme *c*-type cytochrome pathways of Gram-negative bacteria such as *Shewanella* (1, 7), *Geobacter* (1, 7), and the Gram-positive bacterium *Thermincola potens* (16).

Under the electrode respiration condition, transcription and expression of heme synthesis-related genes (*hemY*, *hemH*, *hemE*, *hemL*, and *hemA*) and two *c*-type cytochrome genes (*T479_RS06590* and *T479_RS20980*) in GY32 were significantly upregulated compared with the aerobic condition (Fig. 2; Table 1). The other four *c*-type cytochrome genes (*T479_RS14495*, *T479_RS07305*, *T479_RS18555*, and *T479_RS04160*) were significantly upregulated at the transcriptional level (Table S1). These results indicate that *c*-type cytochromes were particularly needed in the EET of GY32. The Cell-PLoc 2.0 predicted that the *c*-type cytochromes encoded by *T479_RS06590* and *T479_RS20980* were located in the inner membrane and periplasm, respectively (34). Due to the lack of outer membrane, some periplasmic proteins of Gram-positive bacteria can diffuse across the porous cell wall to the cellular outer surface or culture medium (35). Thus, it is possible that the periplasmic *c*-type cytochrome encoded by *T479_RS20980* may participate in the EET by donating electrons to the redox mediators (or conductive nanowires) or diffusing out to the electrode surface.

**Sulfur and cysteine transports and metabolisms.** There were significant differences in sulfur metabolism between electrode respiration and aerobic conditions (Fig. 2; Table S1). In the sulfur metabolism, the proteins of sulfate adenylyltransferase (Sat, converting sulfate to adenosine 5′-phosphosulfate) and sulfite reductase (CysI, converting sulfite to sulfide)

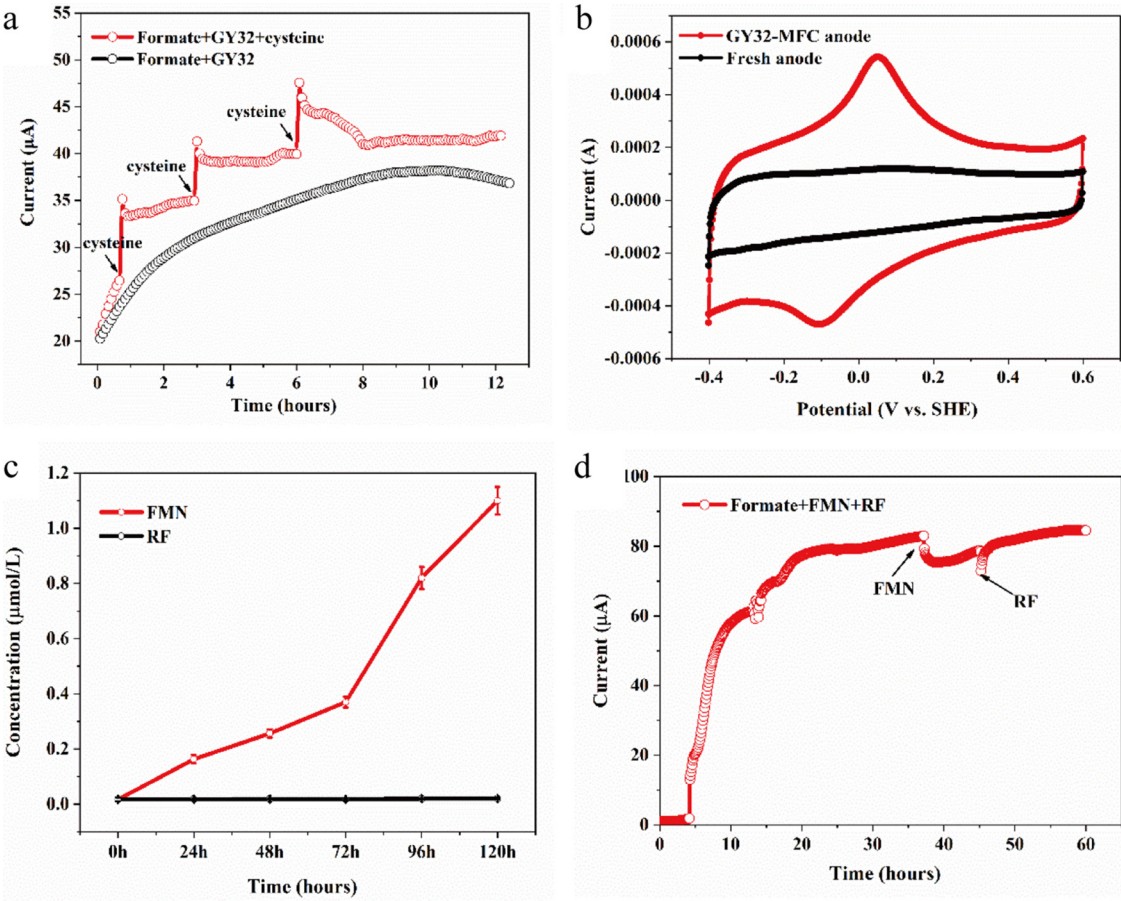

**FIG 4** Bioelectrochemical roles of cysteine and flavins in the current generation of GY32. (a) Cysteine addition increased the current generation of MFC. (b) The anode CV profile of GY32 MFC during current generation. (c) Flavins in the suspension of MFC anode chamber. (d) The effect of flavins on the current generation of GY32. FMN, flavin mononucleotide; RF, riboflavin.

had higher abundance during the respiration with electrode than with oxygen. The expression of Sat under electrode respiration was 5.9 times higher than that under aerobic conditions and was one of the highest in the proteome. In the cysteine metabolism, serine acetyltransferase (CysE, converting L-serine and acetyl-CoA to o-acetyl-serine) was significantly upregulated in electrode respiration of GY32. Sulfide and o-acetyl-serine can be used to generate cysteine. In addition, the formation of sulfide can inhibit the degradation of cysteine (36). These results indicate that cysteine synthesis was enhanced during electrode respiration of GY32 compared with aerobic respiration. After being synthesized, cysteine can be transported across the membrane to the outside via cysteine transferase (a Bcr-family protein), and the extracellular cystine can be transferred back to the cytoplasm by symporter TcyP and ATP binding cassette

**TABLE 1** Transcription and expression results of related genes of *c*-type cytochrome metabolism in GY32

| Gene ID | Gene | Product | Transcriptome | | Proteome | |
|---|---|---|---|---|---|---|
| | | | Log$_2$ fold change | *P* value | Log$_2$ fold change | *P* value |
| T479_RS20160 | *hemY* | Protoporphyrinogen oxidase | 1.58 | 0.00 | 0.63 | 0.00 |
| T479_RS18655 | *hemH* | Ferrochelatase | 2.45 | 0.00 | 0.61 | 0.00 |
| T479_RS18660 | *hemE* | Uroporphyrinogen decarboxylase | 3.98 | 0.00 | 0.25 | 0.45 |
| T479_RS19380 | *hemL* | Glutamate-1-semialdehyde aminotransferase | 2.29 | 0.00 | 0.07 | 0.64 |
| T479_RS15785 | *hemA* | Glutamyl-tRNA reductase | 1.92 | 0.00 | 1.11 | 0.63 |
| T479_RS14675 | *hemN* | Coproporphyrinogen III oxidase | 3.44 | 0.00 | −0.06 | 0.06 |
| T479_RS06590 | | *c*-type cytochrome | 2.60 | 0.00 | 0.88 | 0.01 |
| T479_RS20980 | | class I cytochrome *c* | 5.61 | 0.00 | 0.62 | 0.00 |

transporters TcyK, similar to that in *Bacillus* (37). The cell wall of Gram-positive bacteria is a cross-linked structure of peptidoglycan with a large number of pores, which can allow low-weight molecules such as cysteine or some proteins to pass through (15).

To analyze the cysteine concentration in anodic culture, a cysteine-specific fluorescent sensor (NTPC) was synthesized according to a recent report (38). The NTPC solution with or without cysteine showed no fluorescence under sunlight. When exposed to UV light, fluorescence was observed in the NTPC solution with cysteine but not in the solution without cysteine (Fig. 3). Moreover, NTPC fluorescence intensity showed a good linear relationship with cysteine concentration ranging from 0 to 25 $\mu$M (Fig. S3b, c). During current generation, the cysteine concentration in the anodic culture increased from 0 to 10.8 $\mu$M (Fig. 3). The results indicate that GY32 synthesizes and secretes cysteine into the culture medium. Cysteine, as a small molecule with the sulfhydryl group, can easily form a redox couple of cysteine/cystine in the natural environment. Moreover, during the electrode respiration of GY32, a thiol-disulfide oxidoreductase (YkuV) was significantly upregulated compared with aerobic respiration (Table S1). YkuV is a periplasmic protein and has been reported to catalyze the redox reaction of the cysteine/cysteine couple in *B. subtilis* (39). These results indicate a possibility that GY32 can generate and use cysteine as an electron mediator in the EET to the electrode.

**Cysteine mediated EET of strain GY32.** It has been reported that artificially added cystine/cysteine can improve the EET of some Gram-negative electroactive microorganisms, as well as redox interaction between *G. sulfurreducens* and *Wolinella succinogenes* (40). In order to verify the possible electron mediator role of cysteine in GY32 electrode respiration, we added it to the anode chamber during the initial stage of the current generation (2.0 $\mu$M for three times), and the current generation could be improved correspondingly (Fig. 4a). These results indicate that cysteine added early in the growth phase can enhance the electron transport between the GY32 cell and the electrode.

MFC anode cyclic voltammetry (CV) scanning showed a couple of redox peaks with a midpotential at −40 mV (Fig. 4b), which is close to the previously reported redox potential of cysteine. It has been reported that the cysteine residue in the membrane-anchored sensor kinase ArcB has a redox potential of −41 mV (41). Moreover, the redox potential of cysteine was determined to be −46 mV at the concentration of 4 $\mu$M (42). The electrochemical activity of cysteine makes it a feasible electron mediator to mediate the EET between GY32 and anode. It should be noted that, although we showed evidence that GY32 could secrete and use cysteine as the electron mediator in EET to electrodes, participation of other untested electron mediators cannot be ruled out.

**The cysteine-mediated EET pathway of strain GY32 and implications.** Our results suggest that strain GY32 mainly relies on secreting electron mediators for electrode respiration, and it can secrete and use cysteine as an electron mediator. Assuming the redox potential of the cysteine/cystine couple in the anodic culture was −40 mV, it would be thermodynamically favorable to deliver electrons from many redox proteins or coenzymes to the anode (41). Our recent results showed that the membrane-bound *c*-type cytochromes of GY32 have a midpotential of −178 mV (43). Therefore, the cysteine/cystine couple may be reduced by these *c*-type cytochromes in strain GY32. Moreover, it has been suggested that cysteine could be reduced by some coenzymes with more negative redox potentials, such as NADH (−320 mV), and menaquinones (−74 mV) (Fig. 5) (41, 44). In addition, during the electrode respiration of GY32, the highly expressed thiol-disulfide oxidoreductase (YkuV, redox potential −332 mV) indicates another pathway for the cysteine-mediated EET of GY32 (39). However, whether the YkuV can interact with the *c*-type cytochromes or conductive nanowires remains unknown.

Thiols are ubiquitous in natural environments (45, 46). Cysteine, as a thiol substance, can chemically reduce iron and manganese oxides in sediment (47), thus promoting the redox processes of elements such as S, Fe, and Mn. The biological generation of cysteine, as well as flavins, is an important strategy of bacteria in sensing and adapting to environmental redox conditions (48). Moreover, the redox couple of cysteine/cystine can be used as an electron mediator by some Gram-negative microorganisms in EET and interspecies electron transfer (40, 49, 50). Considering the extremely long cell shape (up to 1 mm) and cysteine/flavins

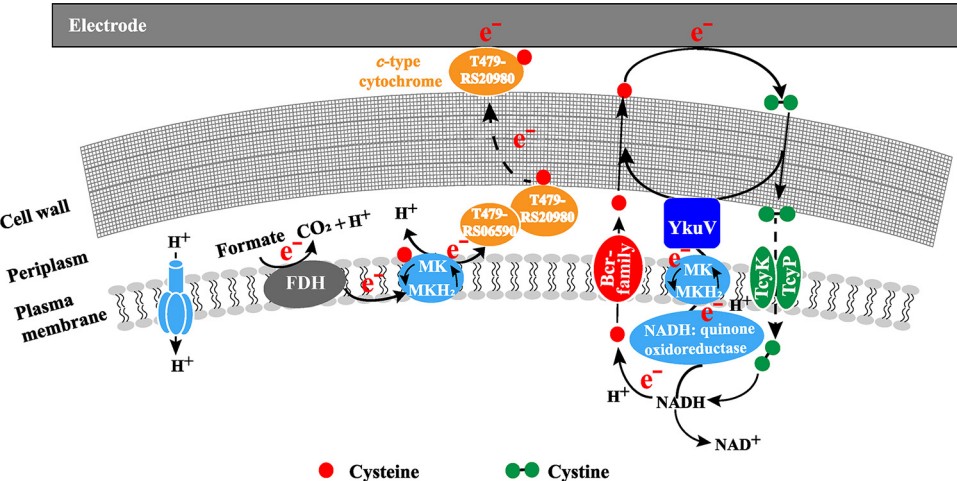

**FIG 5** Proposed EET pathways of *L. varians* GY32. NADH: quinone oxidoreductase (T479_RS01205, T479_RS17800), MK (menaquinones), YkuV (thiol-disulfide oxidoreductase: T479_RS03930), Bcr-family protein (cysteine transferase: T479_RS06160, T479_RS07610, T479_RS05520), TcyP (L-cystine transport protein: T479_RS01130), TcyK (L-cystine-binding protein: T479_RS16540).

generation of *L. varians* GY32, the presence of a single *L. varians* GY32 cell may promote the surrounding EET processes and shape the microbial composition in a much bigger area. It has been reported that the flavins generated by *S. oneidensis* MR-1 can deliver electrons to metals at a distant of 60 $\mu$m (51). Assuming the same functional distance of cysteine, the functional sphere of a single *L. varians* GY32 cell could be 0.01 mm³, which would be visible to the naked eyes and contain millions of short bacteria cells (with a volume of 2 to 10 $\mu$m³ per cell). Therefore, the cysteine/flavin-secretion capability of *L. varians* GY32 further suggests a unique role of this filamentous Gram-positive bacterium in bioelectrochemical systems or natural environments.

**Conclusions.** In this study, we demonstrated that the electroactive filamentous Gram-positive bacterium *L. varians* GY32 had a wide electron donor spectrum in electrode respiration. It can secrete cysteine as an electron mediator for EET. The expression of *c*-type cytochromes and a thiol-disulfide oxidoreductase (YkuV) were upregulated in electrode-respiration and might donate electrons to cysteine in the EET process. The cysteine-mediated EET in a Gram-positive bacterium was discovered for the first time here, which provides a new perspective for understanding the microbial EET networks in bioelectrochemical systems and natural environments.

## MATERIALS AND METHODS

**Bacterial cultivation.** *L. varians* GY32 was isolated from the sediment of the Lianjiang river, which is contaminated with electronic wastes (Guiyu, China) (52). *L. varians* GY32 was cultivated by transferring a single clone to a 5-mL LB medium (10 g/L peptone, 5 g/L yeast extract, 5 g/L NaCl) for 12 h. Then, 2.5 mL of the resulting culture was added to a 250-mL conical flask containing 100-mL LB medium and incubated in a shaker (180 rpm, 30°C) overnight. The cells were harvested in the middle of the exponential growth phase (approximately 10 h) by centrifugation, washed twice, and resuspended with phosphate buffer (0.1 M, pH 7.2) prior to MFC inoculation. For the MFC cultivation, the bacterial cells were inoculated into the anodic medium with 10 mM of either formate, acetate, lactate, or amino acids as electron donors (12.8 g/L of Na$_2$HPO$_4$, 3 g/L of KH$_2$PO$_4$, 0.5 g/L of NaCl, 1.0 g/L of NH$_4$Cl, 0.5 g/L yeast extract, pH 6.8) in each MFC for current generation at an initial bacterial cell density (OD$_{600}$) of 0.05. To test the feasibility of amino acids as potential electron donors, 20 types of amino acids tested in MFC include L-histidine (His), L-glycine (Gly), L-alanine (Ala), L-lysine (Lys), L-cysteine (Cys), L-serine (Ser), L-glutamate (Glu), L-glutamine (Gln), L-leucine (Leu), L-isoleucine (Ile), L-threonine (Thr), L-aspartic acid (Asp), L-asparagine (Asn), L-tryptophan (Try), L-valine (Val), L-arginine (Arg), L-tyrosine (Tyr), L-methionine (Met), L-phenylalanine (Phe), and L-proline (Pro) was added to the anode chambers individually at a final concentration of 1 mmol/L.

**MFC assembly and operation.** Dual-chamber glass MFCs were assembled as previously described (53). Briefly, plain graphite plates (2 cm × 3 cm × 0.1 cm) were used as anodes and cathodes. When needed, an Ag/AgCl electrode (0.198 V versus standard hydrogen electrode [SHE]) was used as a reference electrode for each anode. The anode and cathode chambers were separated with a piece of Nafion115 membrane (12.56 cm²). After assembly and sterilization (115°C for 30 min), the anode chamber (120 mL) was filled with 100 mL of the medium as described above (pH 6.8). Each cathode chamber was filled with

100 mL sterilized phosphate-buffered saline solution (PBS, pH 7.2) containing 50 mM potassium ferricyanide. After inoculation, the anodic medium was flushed with nitrogen for 15 min to achieve an anaerobic condition and then incubated with a sealed cap.

The anode and cathode were connected to a 1,000 $\Omega$ resistor by titanium wires. MFCs were operated at 30°C, and all cultures were prepared in triplicate. The voltage of MFCs under the closed-circuit condition was recorded every 5 min with a multimeter (Keithley 2700, module 7702). Current is calculated by dividing voltage by resistance. To calculate the coulombic efficiency of the formate-fueled MFC, yeast extract in anodic medium was replaced with mineral salts ($MgSO_4 \cdot 7H_2O$ 0.1 g/L, $CaSO_4 \cdot 2H_2O$ 0.05 g/L, $FeCl_3 \cdot 6H_2O$ 0.2 mg/L, $NaMoO_4$ 0.2 mg/L, $MnCl_2 \cdot 4H_2O$ 0.2 mg/L, $CuCl_2 \cdot 2H_2O$ 0.2 mg/L, $ZnSO_4$ 0.2 mg/L, $H_3BO_3$ 0.3 mg/L, and $CoCl_2 \cdot 6H_2O$ 0.4 mg/L). The coulombic efficiency was calculated as described before, and the background current (without formate) was subtracted during the calculation (2).

**Bacteria growth.** Anaerobic growth of strain GY32 cultivation with an electrode as the electron acceptor, the bacteria density in the anode chamber liquid culture was measured by UV-Visible spectrophotometer at a wavelength of 600 nm ($OD_{600}$).

**Detection of formate and flavin.** Formate concentration was measured by high-performance liquid chromatography (HPLC) (LC-20A, SHIMADZU) with a Zorbax SB-C18 (150 mm $\times$ 4.6 mm $\times$ 5 $\mu$m, Agilent) for separation and a UV detector for measurement at 210 nm. Mobile phase: $0.1\%H_3PO_4$ aqueous solution: acetonitrile (V/V) is 98:2.

The flavins concentration was measured by HPLC. Briefly, 2 mL of the culture liquid was filtered via a 0.22 $\mu$m polytetrafluoroethylene filter and then analyzed by high-performance liquid chromatography (LC-20A, SHIMADZU), which is equipped with a Zorbax SB-C18 column (150 mm $\times$ 4.6 mm $\times$ 5 $\mu$m, Agilent) for separation and a fluorescence detector (RF-10AXL, SHIMADZU) for measurement. The excitation wavelength was 450 nm, and the emission wavelength was 520 nm.

**Biofilm observation.** Biofilms in the anode chamber with 100-h current generation and formate as the sole electron donor were stained with the Live/Dead Bac Light staining kit (Life Technologies, L7012) and observed by a confocal laser scanning microscope (CLSM) as described before (53). All assays were performed in triplicate.

**Evaluating the EET strategy of GY32 and the role of electron mediator.** The method for evaluating the possible EET strategy from GY32 to the electrode was conducted as described previously (9). When the current generation of formate-fueled MFC reached the maximum (about 100 h), the biofilm grown-anode was replaced with a sterilized fresh anode and ran for 8 h to evaluate the contribution of biofilm to the current generation. Then, to evaluate the contribution of culture suspension in the current generation, the biofilm grown-anode was returned, and the anode culture liquid (including medium and planktonic cells) was replaced with fresh medium. To test whether the culture suspension contained electron mediators, we collected the supernatant of the suspension by centrifugation at 8,000 $\times$ $g$ for 5 min. Then, 3 mL of the supernatant was added to the anode chamber every 2 h three times. MFC with fresh medium and sterile MFC with supernatant were used as controls for the supernatant-added biofilm-containing MFC.

To verify the possible electron mediator role of cysteine and flavins, cysteine (working concentration 2 $\mu$mol/L) was added to the anolyte every 2 h three times at the initial stage of the current generation by strain GY32. At the same time, the current generation was detected, and the MFCs added with the fresh medium were used as control. Flavins (2.0 $\mu$mol/L FMN and RF) were added to the MFC anodic chamber when the MFC current output was rising. At the same time, the current generation was detected.

**Transcriptomic and proteomic analyses.** Transcriptomes of strain GY32 respiring with either electrode or oxygen were comparatively analyzed. Formate was used as the only electron donor in electrode respiration of strain GY32. The cells were harvested in the middle of the exponential growth phase for transcriptome analysis. All assays were performed in triplicate.

RNA-seq library construction and high-throughput sequencing were performed as previously described (54). The data analysis is briefly described as follows: raw sequence data were filtered to remove reads aligned to the barcode adapter using FASTP (https://github.com/OpenGene/fastp), reads with $\geq$10% unidentified nucleotides, and reads with $>$50% bases having Phred quality scores of $\leq$20 were removed. The clean reads were mapped to the GY32 genome (accession number: CP006837.1) using Bowtie 2 (version 2.2.8) to identify known genes. The gene transcription was calculated using RNA-Seq by expectation maximization (RSEM). The gene expression level was further normalized by using the fragments per kilobase of transcript per million (FPKM) mapped reads method to eliminate the influence of different gene lengths and amount of sequencing data on the calculation of gene expression. The edge R package (http://www.r-project.org/) was used to identify differentially expressed genes (DEGs) across samples with fold changes $\geq$2 and a false discovery rate adjusted P (q value) $<$0.05. DEGs were then subjected to enrichment analysis of gene ontology (GO) function and Kyoto Encyclopedia of Genes and Genomes (KEGG) pathways, and $q$ values were corrected using $<$0.05 as the threshold (55).

**Data-independent acquisition.** Nano-high performance liquid tandem mass spectrometry (HPLC-MS/MS) analysis was used for proteomic quantification. The peptides were analyzed by online nano spray liquid chromatography-tandem mass spectrometry (LC-MS/MS) on an Orbitrap Fusion Lumos coupled to EASY-nL C 1200 system (Thermo Fisher Scientific, MA, USA). A 3-$\mu$L peptide sample was loaded onto the analytical column (Acclaim Pep Map C18, 75 $\mu$m $\times$ 25 cm). The column flow rate was maintained at 200 nL/min with a column temperature of 40°C. The electrospray voltage of 2 kV versus the inlet of the mass spectrometer was used. The mass spectrometer was run under the data-independent acquisition mode and automatically switched between MS and MS/MS mode.

Raw data of data-independent acquisition (DIA) were processed and analyzed by Spectronaut X (Biognosys AG, Switzerland) with default parameters. Retention time prediction type was set to dynamic

iRT. Data extraction was determined by Spectronaut X based on the extensive mass calibration. Spectronaut Pulsar X will determine the ideal extraction window dynamically depending on iRT calibration and gradient stability. Q value false discovery rate (FDR) cutoff on precursor and protein level was applied 1%. Decoy generation was set to mutated which is similar to scrambled but only applied a random number of AA position swaps (min = 2, max = length/2). All selected precursors passing the filters were used for quantification. The average top 3 filtered peptides, which passed the 1% Q value cutoff, were used to calculate the major group quantities. After Student's t-test, differently expressed proteins were filtered if their Q value <0.05 and $|\log_2$ (fold change) $|\geq$0.58. Proteins were annotated against GO, KEGG, and the Cluster of Orthologous Groups (COG/KOG) database to obtain their functions. Significant GO functions and KEGG pathways were examined within differentially expressed proteins with qvalue ≤ 0.05 (56, 57).

The locations of encoded proteins were predicted using Cell-Ploc 2.0 with Gneg-mPLoc (34).

**Determination of cysteine with the fluorescent sensor.** The fluorescent sensor NTPC was synthesized as previously described (38), and the brief synthesis principle of NTPC is shown in Fig. S3a. The nuclear magnetic resonance (NMR) and high-resolution mass spectrometer (HRMS) were used to identify the intermediate products. The fluorescence intensity of NTPC in the presence of cysteine was determined with the maximum excitation wavelength $\lambda_{Ex}$ = 547 nm and the maximum emission wavelength $\lambda_{Em}$ = 569 nm (Fig. S4b). Moreover, the fluorescence intensity of NTPC shows a good linear positive correlation with the concentration of cysteine in the range of 0 to 25 $\mu$M (Fig. S4c).

For formate-fueled MFC, the cysteine in the supernatant of MFC during the current generation was detected with the NTPC. The suspension of MFC during the current generation was obtained regularly, and the supernatant was collected by centrifugation (6,000 rpm, 5 min). The fluorescent probe NTPC was mixed with the supernatant and stood for 30 min, and then the fluorescence intensity was measured by a fluorescence spectrometer (FS-45, Perkin Elmer, USA) at the excitation wavelength of 547 nm and the emission wavelength of 569 nm. Cysteine concentration was calculated according to the formula of the linear relationship. All assays were performed in triplicate.

**Cyclic voltammetry.** When the current generation reached the maximum (about 100 h), CV was measured, and the device without strain GY32 was used as a control. Experiments were carried out in a three-electrode system to investigate the electrochemical activity of strain GY32 by an electrochemical workstation (CHI1040C, China), with an anode as the working electrode, a cathode as the counter electrode, and an Ag/AgCl as a reference electrode (53). In brief, the potential increment of the CV test was 1 mV; the scan range was − 0.6V to 0.4V versus Ag/AgCl; the scan rate was 1 mV/s. Data of the third CV scan cycle was presented.

**Data availability.** Raw data have been uploaded to National Center for Biotechnology Information (NCBI) with an accession number of GSE165754.

## SUPPLEMENTAL MATERIAL

Supplemental material is available online only.
**SUPPLEMENTAL FILE 1**, PDF file, 1.3 MB.

## ACKNOWLEDGMENTS

This study was supported by the Key-Area Research and Development Program of Guangdong Province (2020B1111380003), National Natural Science Foundation of China (91851202 and 31970110), and Science and Technology Project of Guangdong Academy of Sciences (2022GDASZH-2022010203, 2020GDASYL-20200103036).

All authors conceived the study and revised the manuscript. Y.Y., M.X., and G.K. designed the experiments. G.K. performed the experiments. Y.L. and F.L. synthesized fluorescent probes (NTPC). G.K. and D.S. analyzed the data. G.K. and Y.Y. wrote the manuscript. M.D. revised the manuscript.

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
