## [Reviewer comments · Microbiology Spectrum]

Microbiology Spectrum

Cysteine-mediated extracellular electron transfer of *Lysinibacillus varians* GY32

Guannan Kong, Yonggang Yang, Yeshen Luo, Fei Liu, Da Song, Guoping Sun, Daobo Li, Jun Guo, Meijun Dong, and Meiyong Xu

Corresponding Author(s): Yonggang Yang, Guangdong Institute of Microbiology

Review Timeline:

Submission Date:	July 27, 2022
Editorial Decision:	September 2, 2022
Revision Received:	September 30, 2022
Accepted:	October 4, 2022

Editor: Daniel Bond

Reviewer(s): The reviewers have opted to remain anonymous.

Transaction Report:

DOI: <https://doi.org/10.1128/spectrum.02798-22>

September 2, 2022

Dr. Yonggang Yang
Guangdong Institute of Microbiology
100#, Xianliezhong Road, Guangzhou, China
Guangzhou, Guangdong
China

Re: Spectrum02798-22 (Cysteine-mediated extracellular electron transfer of *Lysinibacillus varians* GY32)

Dear Dr. Yonggang Yang:

Thank you for submitting your manuscript to Microbiology Spectrum. This manuscript presents a number of interesting new observations relating to electron transfer out of *Lysinibacillus*, including identification of a new electron shuttle using a novel detection strategy.

Many initial reviewer concerns related to correctly reporting the physiological conditions and accounting for low coulombic recovery. Most of these have been addressed during a prior cycle of responses.

Based on reviewer input, only a few physiological and experimental issues must be addressed prior to publication, and the editor's decision is Modifications.

Link Not Available

Sincerely,

Daniel Bond

Journals Department
Reviewer comments:

1. Regarding the claim that flavin is not used as shuttles but cysteine is; these experiments were not conducted in a way that allows this conclusion in the abstract. Cystine was added in multiple doses very early in growth (as early as 45 minutes after inoculation). Each flavin were added as a single dose 40 hours after inoculation. Early in growth, there was likely a lot of oxygen,

and no accumulated shuttles, while later, the plateau had already been reached so there is no room for improvement. That said, the authors give a hypothesis for why cystine could be used compared to flavins in terms of redox potential and genomics and putting this data forward can enable cause further testing.

To allow readers to make their own conclusions, please make these two changes.

A) Move Figure S3 to be part B of Figure 4 so readers can see the experiments side by side.

B) Alter the text of the abstract to reflect what can be concluded about cysteine, without the unproven conclusion regarding flavins;

"Fluorescent sensor and electrochemical analyses demonstrate that strain GY32 can secrete cysteine and flavins. Cysteine added shortly after inoculation into microbial fuel cells accelerated EET, showing cysteine is a new endogenous electron mediator of Gram-positive bacteria...

Minor changes:

Line 84; "could perform more efficient EET..." change to "could perform more rapid EET..." Efficiency reflects yield, these experiments involve rates.

Line 95; "performed relatively high ETT efficiency" change to "performed relatively high ETT rates"

Line 107: after "...was calculated to be 18%". Include a statement (from responses to reviewers) such as "tests found oxygen present in reactors that increased over time", or "The lower coulombic yields were not surprising, as measurements found detectable oxygen diffusion into the two-chambered fuel cells similar to other experiments with facultative anaerobes..."

Good reactors with *Geobacter* have to be well sealed as this anaerobe will die with too much oxygen, thus they achieve close to 95% coulombic efficiency. Standard *Shewanella* reactors, especially 2-chambered ones, have low efficiency as they prefer to use oxygen. See

'Komal Joshi et al., "Preventing Hydrogen Disposal Increases Electrode Utilization Efficiency by *Shewanella oneidensis*," *Frontiers in Energy Research* 7 (2019): 95, <https://doi.org/10.3389/fenrg.2019.00095>.' for more discussion regarding oxygen.

Line 181: indicate the timing of flavin addition "...MFC anodic chamber of GY32 after 40 h of growth..."

Line 183; "...showed that GY32 synthesized flavins but did not benefit from higher levels after growth for 40 h."

Line 248; "These results indicate that cysteine added early in the growth phase can enhance..."

Figure 5; Remove the arbitrary 'Nanowire' from the figure. In light of recent findings that most structures presumed to be nanowires from crude imaging are actually other proteins such as cytochromes or even DNA, such speculation without identification of the object is discouraged.

Staff Comments:

Preparing Revision Guidelines

Please return the manuscript within 60 days; if you cannot complete the modification within this time period, please contact me. If you do not wish to modify the manuscript and prefer to submit it to another journal, please notify me of your decision immediately so that the manuscript may be formally withdrawn from consideration by Microbiology Spectrum.

Point-by-point Responses to the Reviewer comments

1. Regarding the claim that flavin is not used as shuttles but cysteine is; these experiments were not conducted in a way that allows this conclusion in the abstract. Cystine was added in multiple doses very early in growth (as early as 45 minutes after inoculation). Each flavin were added as a single dose 40 hours after inoculation. Early in growth, there was likely a lot of oxygen, and no accumulated shuttles, while later, the plateau had already been reached so there is no room for improvement. That said, the authors give a hypothesis for why cystine could be used compared to flavins in terms of redox potential and genomics and putting this data forward can enable cause further testing.

Response: Thank you very much for your comments and suggestions. We agree with you and have revised the manuscript accordingly.

To allow readers to make their own conclusions, please make these two changes.

A) Move Figure S3 to be part B of Figure 4 so readers can see the experiments side by side.

Response: Thank you. We have move figure S3a and S3b to be figure 4c and 4d in the revised manuscript. The captions have also been revised accordingly.

B) Alter the text of the abstract to reflect what can be concluded about cysteine, without the unproven conclusion regarding flavins:

"Fluorescent sensor and electrochemical analyses demonstrate that strain GY32 can secrete cysteine and flavins. Cysteine added shortly after inoculation into microbial fuel cells accelerated EET, showing cysteine is a new endogenous electron mediator of Gram-positive bacteria..."

Response: Accepted. The abstract has been revised according to your suggestion (line 23-27): "Fluorescent sensor and electrochemical analyses demonstrate that strain GY32 can secrete cysteine and flavins. Cysteine added shortly after inoculation into microbial fuel cells accelerated EET, showing cysteine is a new endogenous electron mediator of Gram-positive bacteria, which provides novel information to understand the EET networks in natural environments."

Minor changes:

Line 84; "could perform more efficient EET..." change to "could perform more rapid EET..." Efficiency reflects yield, these experiments involve rates.

Response: Accepted. "could perform more efficient EET..." has been changed to "could perform more rapid EET..." (line 84).

Line 95; "performed relatively high ETT efficiency" change to "performed relatively high ETT rates"

Response: Accepted. "performed relatively high ETT efficiency" has been changed to "performed relatively high ETT rates" (line 95).

Line 107: after "...was calculated to be 18%". Include a statement (from responses to reviewers) such as "tests found oxygen present in reactors that increased over time", or "The lower coulombic yields were not surprising, as measurements found detectable oxygen diffusion into the two-chambered fuel cells similar to other experiments with facultative anaerobes..."

Response: Accepted. The following discussions has been added (line 106-108): "The lower Coulombic efficiency was not surprising, as detectable oxygen (0.1 μ M) diffused into the anode chamber similar to other experiments with facultative anaerobes (25)."

Good reactors with Geobacter have to be well sealed as this anaerobe will die with too much oxygen, thus they achieve close to 95% coulombic efficiency. Standard Shewanella reactors, especially 2-chambered ones, have low efficiency as they prefer to use oxygen. See

'Komal Joshi et al., "Preventing Hydrogen Disposal Increases Electrode Utilization Efficiency by Shewanella oneidensis," Frontiers in Energy Research 7 (2019): 95, <https://doi.org/10.3389/fenrg.2019.00095>.' for more discussion regarding oxygen.

Response: Agree. We have learned the ingenious experiments in the article suggested by the reviewer. GY32 has Ni-Fe hydrogenases (WP_025219112, WP_025219113).

We further discussed the possible effect of the hydrogenases of GY32 on the Coulombic efficiency (in line 108-112),"Moreover, it has been reported the hydrogen generation caused by hydrogenases of S. oneidensis MR-1 could account for 38% of electron loss in the current generation (25). The Ni-Fe hydrogenases of GY32 (WP_025219112, WP_025219113) may also divert electron transfer from current generation to hydrogen production.) ". The suggested article (Ref. 25) has been cited in the revised manuscript.

Line 181: indicate the timing of flavin addition "...MFC anodic chamber of GY32 after 40 h of growth..."

Response: Accepted. This sentence has been changed to (line 186-187): "...we added flavins (2.0 μ M FMN and RF) to the MFC anodic chamber of GY32 after 40 h of growth and found that they could not increase the current generation (Fig. 4d)."

Line 183; "...showed that GY32 synthesized flavins but did not benefit from higher levels after growth for 40 h."

Response: Accepted. This sentence has been changed to (line 188-189): "These results showed that GY32 synthesized flavins but did not benefit from higher levels after growth for 40 h..."

Line 248; "These results indicate that cysteine added early in the growth phase can enhance..."

Response: Accepted. This sentence has been changed to (line 253-254): “These results indicate that cysteine added early in the growth phase can enhance the electron transport between the GY32 cell and the electrode.”

Figure 5; Remove the arbitrary 'Nanowire' from the figure. In light of recent findings that most structures presumed to be nanowires from crude imaging are actually other proteins such as cytochromes or even DNA, such speculation without identification of the object is discouraged.

Response: Agree. The “Nanowire” has been removed from Figure 5.

October 4, 2022

Dr. Yonggang Yang
Guangdong Institute of Microbiology
100#, Xianliezhong Road, Guangzhou, China
Guangzhou, Guangdong
China

Re: Spectrum02798-22R1 (Cysteine-mediated extracellular electron transfer of *Lysinibacillus varians* GY32)

Dear Dr. Yonggang Yang:

I am pleased to inform you that your manuscript has been accepted, and I am forwarding it to the ASM Journals Department for publication. You will be notified when your proofs are ready to be viewed.

Sincerely,

Daniel Bond
Editor, Microbiology Spectrum
